# Characterization of the Internal Stress Evolution of an EB-PVD Thermal Barrier Coating during a Long-Term Thermal Cycling

**DOI:** 10.3390/ma16072910

**Published:** 2023-04-06

**Authors:** Zhen Zhen, Chuan Qu, Donghui Fu

**Affiliations:** 1BAIMTEC Material Co., Ltd., Beijing 100094, China; 2Department of Mechanics, Tianjin University, Tianjin 300350, China

**Keywords:** thermal barrier coating, electron beam physical vapour deposition (EB-PVD), spectroscopy, stress evolution

## Abstract

Electron beam physical vapour deposition (EB-PVD) technology is a standard industrial method for the preparation of a thermal barrier coating (TBC) deposition on aeroengines. The internal stress of EB-PVD TBCs, including stress inside the top coating (TC) and thermal oxidation stress during long-term service is one of the key reasons for thermal barrier failures. However, research on the synergistic characterization of the internal stress of EB-PVD TBCs is still lacking. In this work, the stress inside the TC layer and the thermal oxidation stress of EB-PVD TBC during long-term thermal cycles were synergistically detected, combining Cr^3+^-PLPS and THz-TDS technologies. Based on a self-built THz-TDS system, stress-THz coefficients *c*_1_ and *c*_2_ of the EB-PVD TBC, which are the core parameters for stress characterization, were calibrated for the first time. According to experimental results, the evolution law of the internal stress of the TC layer was similar to that of the TGO stress, which were interrelated and influenced by each other. In addition, the internal stress of the TC layer was less than that of the TGO stress due to the columnar crystal microstructure of EB-PVD TBCs.

## 1. Introduction

With the rapid development of major national projects and equipment industries, such as aerospace and thermal power energy, there is an increasingly urgent demand for functional coatings with excellent resistance to high temperature, wear and corrosion [1,2,3,4,5,6]. Thermal barrier coating (TBC), a typical functional coating, plays an important role in the aerospace industry due to its excellent heat insulation performance and high thermal stability, increasing the operating temperature of metallic engine components such as turbine blades in the hot section of an engine [7,8]. TBCs have a protective multilayer structure, which is mainly composed of the top coat (TC), the bond coat (BC) and the superalloy [9]. Thermal growth oxide (TGO), which is mainly composed of *α*-Al_2_O_3_, may grow at the interface between the TC and the BC under high-temperatures. TBCs are mainly prepared by two methods; one is electron beam physical vapour deposition (EB-PVD), and the other is atmospheric plasma spraying (APS). In particular, EB-PVD has become a standard method of TC layer deposition in the aeroengine industry [10,11]. In the EB-PVD process, the target material heated by a high-energy electron beam is deposited on the surface of the metal substrate in a high-vacuum condition [12]. EB-PVD TBCs with a columnar structure have a much higher degree of strain tolerance than those prepared by plasma spray. Moreover, an EB-PVD coating also has a smooth surface with good erosion durability and good adherence to a smooth substrate.

The stress introduced inside TBCs during their manufacturing and service is one of the important factors for the spalling failure of TBCs. TBCs typically have a multilayer heterostructure. Hence, due to the difference of physical and chemical parameters and thermodynamic performances of each layer in TBCs, a complex stress field was introduced into TBCs [13,14,15,16,17]. For instance, the thermal mismatch between layers and the microstructural evolution of the TC layer during the thermal service life introduce the internal stress of the TC layer [18,19,20]. Meanwhile, an interfacial stress is generated at the TC/BC interface, owing to the irregular growth and thickening of the TGO. A large number of microcracks will germinate due to the internal and interfacial stresses. During subsequent service, microcracks will expand and merge along the interface region, eventually inducing the peeling and failure of the TBC system, which seriously threatens the operational safety of aeroengines [21,22,23,24,25]. Hence, non-destructive and accurate characterization of the TBC internal and interfacial stresses is essential to precisely evaluate the operational safety and service life of aeroengines.

Many non-destructive experimental methods have been proposed to characterize stress inside TBCs, such as photoluminescence piezospectroscopy (PLPS), micro-Raman spectroscopy (μRS), X-ray diffraction (XRD) and terahertz time-domain spectroscopy (THz-TDS). However, μRS and XRD techniques are confined only to the characterization of TBC surface stress, and they have long experiment time [26,27,28].

PLPS technology has a wide application in stress characterization and debonding detection of TBCs [29,30,31,32,33,34]. Cr^3+^ PLPS technology has been widely used to characterize the TGO stress [35,36,37,38,39]. The measurement principle of Cr^3+^ PLPS technology is that the stress is linearly correlated with the fluorescence spectral frequency shift of the peak position of the Cr^3+^ fluorescence spectrum. In 1993, the quantitative relationship between the stress and the frequency shift of the Cr^3+^ photoluminescence piezospectroscopy of crystalline materials was found by Ma and Clarke [40], and it became the basis for research on the stress measurement of ceramics using PLPS technology. In 2017, the dynamic evolution of average thermal oxidation stress in the TGO during thermal cycles was characterized by Lima et al., who established the interfacial damage model based on stress evolution [41]. Jiang et al. [34] and Sridharan et al. [42] characterized the TGO stress fields of APS TBCs and EB-PVD TBCs, respectively, under different thermal cycles.

THz-TDS technology has great potential for measuring the stress inside the TC layer in TBCs [43,44,45,46] because THz waves can transmit through most non-metallic materials and present the properties of stress-induced birefringence. Based on the principle of photoelasticity, the stress inside the TC layer can be characterized by the transmitted THz spectra [47,48,49]. In 2007, based on a polarization-sensitive THz system, the stress inside polyethylene (PE) was qualitative characterize by Tsuguhiro et al. [50]. In 2016, based on polarization-sensitive THz waves, stress inside polytetrafluoroethylene (PTFE) was quantitatively characterized by Wang et al. [51]. Schemmel et al. demonstrated that THz-TDS technology can be used to quantitatively characterize the stress inside the TC layer, based on who observed the birefringence phenomenon of the sintered ceramics [52]. Recently, the stress-THz coefficients *c*_1_ and *c*_2_ of the sintered APS TC layer were calibrated by Wang et al., and were then used for the measurement of TBC internal stress during the initial thermal cycles [53,54]. However, the stress-THz coefficients of the TC layer vary greatly due to differences in the microstructure of sintered ceramics and EB-PVD TBC. Furthermore, there is a lack of experimental research on the calibration of the stress-THz coefficients of the EB-PVD TBC, which is a significant parameter in THz-stress analysis.

Existing studies have shown that the thermal mismatch stress inside the TC layer and the TGO layer is the main reason for the peeling and failure of the TBC system [55]. In addition, volume expansion induced by the phase transition of the TC layer will accelerate the initiation and propagation of cracks [56]. However, among the existing studies on stress characterizations of EB-PVD TBCs mainly focus on the characterization of thermal oxidation stress but ignore the characterization of the internal stress of the TC layer and the interaction between the TC layer and the TGO layer. There is a lack of synergistic characterization of the stress inside the TC layer and thermal oxidation stress in TGOs (viz. between the TC and the BC coatings).

In this work, the stress evolution of the internal stresses of the TC layer and TGO of EB-PVD TBCs during long-term thermal cycling was synergistic characterized in this work by combining PLPS and THZ-TDS. The stress-THz coefficients *c*_1_ and *c*_2_ of the EB-PVD TC layers in compression states were calibrated for the first time using a self-built reflection-type THz-TDS system. After undergoing different thermal cycles (from 100 to 7000), the spectral characteristics of the EB-PVD TBC specimens were detected by Cr^3+^ PLPS and THz-TDS to quantify the thermal oxidation stress in TGO and the stress inside the TC layer, respectively. Based on the experimental results, the stress evolution mechanism of the internal stresses of the TC layer and the TGO of EB-PVD TBCs during long-term thermal cycling is discussed.

## 2. Materials and Experiments

### 2.1. Specimen Preparation

In this work, the substrate material was single crystal NiCrAlYHf, which was cut into slices of 30 × 10 × 1.5 mm. After that, 100 μm thick TC layers were deposited on both sides of the substrate using the EB-PVD technique. The electron beam evaporation process was carried out in a coating chamber. The target material was heated, and the resulting vapor condensed on the NiCrAlYHf substrates. The NiCrAlYHf substrate acts as a BC layer. All specimens were coated at the same time in a single batch. After spraying, the specimens were cooled to room temperature. The TC layer was made of 8 wt.% yttrium oxide partially stabilized zirconia (8YSZ) powder. The structure of the specimens is shown in Figure 1a.

The specimens underwent different thermal cycles using a high-temperature muffle furnace. The numbers of the thermal cycles were 0, 100, 500, 1000, 2000, 5000 and 7000. These specimens were named S1 to S7, respectively. Each thermal cycle included a 5 min heating period from indoor temperature to a maximum temperature of 1100 °C and a 5 min heat preservation period at 1100 °C, followed by a 5 min forced air quench from 1100 °C to indoor temperature. The maximum temperature of each thermal cycle was 1100 °C. Figure 1b shows all specimens used in this work.

### 2.2. Experiments

Figure 2 shows a reflection-type THz-TDS system with an all-fibre THz time-domain spectrometer used in this work. THz waves were generated and detected by two THz antennas, respectively, which can produce 0.2~1.40 THz reliable THz pulse. The femtosecond laser wavelength is 1550 nm. A convex lens was used to concentrate THz waves, and the diameter of the light spot was 8 mm. One polarizer was placed between the convex lenses of the incident light path, and another polarizer was placed between the two convex lenses of the detection light path. The incident angle of the THz wave was 45°.

The THz-TDS system described above was used to calibration the stress-THz coefficients, *c*_1_ and *c*_2_, of the as-deposited specimens. To measure coefficient *c*_1_, the loading direction was established in the same plane as the polarization direction of the polarizer. The axis of the specimen coincided with the loading direction. The specimens were uniaxial loaded with a maximum load of 1400 N and a step size of 100 N. The THz reflection signal is collected after each loading by averaging 100 in situ collections. The acquisition length of the THz data was 40 ps, and the data resolution was 0.02 ps. For the stress-THz coefficient *c*_2_, the loading direction was rotated 90° perpendicular to the polarization direction of the polarizer. The stress-THz coefficient *c*_2_ was obtained by the same method as *c*_1_.

We collected the THz time-domain spectra of S1 to S7 to measure the internal stress of EB-PVD TBCs. A miniature hexapod microrobot (H-811. I2, Physik Instrumente Ltd., Karlsruhe, Germany) was used for specimen movement. The scanning length was 12 mm and the step size was 2 mm; hence, in this manner, the spectral data of 7 sampling spots were obtained. The experimental parameters were similar to those used in the calibration tests above.

The Cr^3+^ fluorescence spectra of S1 to S7 were measured to characterize the TGO stress. A micro-Raman/PL spectroscope (Raman-RTS, Pioneer Technology Ltd., Hong Kong, China) was used to excite and collect fluorescence data, as shown in Figure 3. In this spectroscope, a 532-nm laser and an 1800-L/mm grating were used. The exposure time was 2 s, and the spectral range was 690~700 nm. The spectral resolution of the spectroscope was approximately 0.01 nm. A 20× objective lens was chosen to focus the laser on the surface of the specimen. When the laser arrived at the TGO, the sampling spot size was approximately 100 μm. A 20 × 7.5 mm rectangular area was detected with a step length of 500 μm; hence, in this manner, the spectral data of 600 sampling spots were obtained.

The thickness of each TC layer was measured using a three-dimensional surface topography instrument (ST400, Nanovea, Irvine, CA, USA). Specimens under different thermal cycles were scanned longitudinally three times. The scan interval was 1.5 mm. The scanning length of the specimens was 30 mm, with a step length of 1 μm.

## 3. Results and Discussion

### 3.1. Characterization of Internal Stress of TBC

#### 3.1.1. Principle of THz Stress Characterization

In opaque materials such as TC layers, the sensitive stress-induced birefringence effect of THz waves has been observed [57,58,59]. The time delay of THz waves when they encounter the material is shifted. Based on small elastic deformation, the changes in the refractive index of each principal stress is linearly correlated with the changes in the stresses. The formula is as follows:(1)$$\{\begin{array}{c} n_{1}-n_{0}=c_{1}\sigma_{1}+c_{2}(\sigma_{2}+\sigma_{3})\\ n_{2}-n_{0}=c_{1}\sigma_{2}+c_{2}(\sigma_{1}+\sigma_{3})\\ n_{3}-n_{0}=c_{1}\sigma_{3}+c_{2}(\sigma_{1}+\sigma_{2}) \end{array}$$
where *σ*_i_ is the principal stress (i = 1, 2, 3), *n*_i_ is the refractive index of the material in the directions of *σ*_i_., *c*_1_ and *c*_2_ are the stress-THz coefficients, and *n*_0_ is the refractive index of the material without loadings.

Considering the thin film structure of a TBC [60], the stress state of the TC layer can be simplified as the in-plane biaxial stress state (*σ*_1_ = *σ*_2_ = *σ*, *σ*_3_ = 0). Equation (1) reduces to
(2)$$\Delta n_{1}=c_{1}\Delta \sigma_{1}+c_{2}\Delta \sigma_{2}=(c_{1}+c_{2})\Delta \sigma_{1}$$

If the TC layer is only subjected to uniaxial compression, viz., *σ*_1_ ≠ 0, while *σ*_2_ = 0, Equation (1) reduces to:(3)$$\{\begin{array}{c} \Delta n_{1}=c_{1}\Delta \sigma_{1}\\ \Delta n_{2}=c_{2}\Delta \sigma_{1} \end{array}$$

Based on the linear fitting slope, the stress-THz coefficients *c*_1_ and *c*_2_ can be calibrated.

A THz wave can penetrate a ceramic film but cannot transmit through a metal material. Based on reflected THz pulses, THz waves are reflected at the specimen surface and the YSZ/BC interface. Figure 4a shows the path of a THz wave incident to a specimen as utilized in this work, where I is the incident THz wave, S is the reflected part of the incident light on the specimen surface, R1 is the THz wave refracted into the YSZ layer and totally reflected firstly on the YSZ/BC interface and then partly refracted into the air at the YSZ/air interface (i.e., the TBC surface), R2 is the part that was totally reflected, firstly on the YSZ/BC interface and then on the YSZ/air interface and reflected secondly on the YSZ/BC interface and then refracted into the air at the YSZ/air interface. Figure 4b gives a typical THz time-domain spectrum, where Δ*t*_1_ is the time delay of the THz wave between S and R1 and Δ*t*_2_ is the time delay of the THz wave between R1 and R2. Figure 4b shows that, generally, Δ*t*_1_ = Δ*t*_2_ = Δ*t*. The specimen refractive index, *n*_TC_, is linearly related to the time delay, Δ*t*. The formula is as follows:
(4)$$n_{\mathrm{TC}}=\frac{c\Delta t\cos\alpha }{2d}$$
where *d* is the specimen thickness, *c* is the velocity of light, and *α* is the angle of refraction. The refractive index of the EB-PVD TBCs was 3.977 [61]. The refraction angle, *α*, was 10.242, based on the refraction law and Equation (4), corresponding to cosα being 0.98406. Cos*α* is considered to be a constant in this study.

Using Equations (2) and (4), the stress inside the TC layer can be expressed as:(5)$$\sigma_{1}=\sigma_{2}=\frac{1}{\left(c_{1}+c_{2}\right)}\cdot \left(\frac{0.49203c\Delta t}{d}-n_{0}\right)$$

#### 3.1.2. Thickness of TC Layer

According to Equation (4), the thickness of the coating, *d*, is one of the essential parameters of THz stress characterization. The surface morphology of each specimen was scanned to measure the thickness of the TC layer. Figure 5 shows the result of S1. During the process of topography scanning, each specimen was scanned longitudinally from one end near the round hole to the other end. A one-dimensional coordinate system was established to describe the position of each sampling spot, with the scanning origin as the coordinate origin and the scanning direction as the positive direction. As shown in Figure 1, there was no coating deposited on the substrate at either end of any specimen. The thickness of the TC layer at each sampling spot was regarded as equal to the height difference between the surface of the TC layer and the free surface of the substrate at the two ends.

As shown in Figure 5, the specimen was thicker in the middle than on either side, showing an uneven distribution. Meanwhile, the results of the three topography scans of the specimens were almost similar, indicating that the thickness of the selected region was evenly distributed in the transverse direction. There were many prominences on the surface of the specimens due to the columnar structure of the EB-PVD TBC. Based on the results shown in Figure 5, the standard deviation of the thickness of the TC layer is 14.47 μm. Due to the large thickness gradients in the edges, the central region of each specimen was selected for THz measurement. The standard deviation of thickness in the selected area (5~25 mm position in Figure 5) was 6.92 μm. Taking the 9 mm position in Figure 5 as the starting spot, THz measurements were performed step by step along the length of the specimen with step lengths of 2 mm. THz data were collected at 7 sampling spots for each specimen. The spot diameter of THz-TDS in this work was approximately 8 mm. Hence, the average thickness inside the area with a diameter of 8 mm was regarded as the TC thickness corresponding to this sampling spot for calculating stress using Equation (5). The thicknesses from the THz measurements at every sampling spot on all specimens are shown in Figure 6.

#### 3.1.3. Calibration of Stress-THz Coefficient

The stress-THz coefficients *c*_1_ and *c*_2_ were calibrated as follows. To measure coefficient *c*_1_ of the TC layer, we collected THz spectra of S1 with different stresses, where the THz polarization direction was consistent with the loading direction. Based on the barycentre method and Equation (4), the refractive indices of S1 with different uniaxial stresses were obtained. In the barycentre method, the barycentric peak was used for stress-THz coefficient measurement. According to the image characteristics and discrete data of the THz time-domain spectra, the barycentre method was used for data processing to avoid the interference of spectral asymmetry and environmental factors on data processing, which improved the stability and reliability of the calibration results. In addition, the specimen thickness for the calibration experiment was 105.15 μm, which corresponded to the position of the specimen at 15 mm in Figure 5. In Figure 7a, the refractive index of S1 increased with an increase in uniaxial compressive stress and had a good linear relationship. The stress-THz coefficient, *c*_1_, of the TC layer under compression loadings was −0.4302 ± 0.0104/GPa. In addition, in most cases, the TC layer of the TBC was under compressive stress state. Hence, the stress-THz coefficient of TBC with compression was important in the actual analysis of the TC layer.

To measure coefficient *c*_2_ of the TC layer, the direction of uniaxial loading was perpendicular to the linear polarization direction of the THz wave. Otherwise, the experimental process and measurement method for measuring the stress-THz coefficient, *c*_2_, were identical to those for *c*_1_. In Figure 7b, the refractive index of S1 increased with an increase in uniaxial compressive stress and showed a good linear relationship. Based on the barycentre method, the stress-THz coefficient, *c*_2_, of the TC layer was −0.1553 ± 0.0073/GPa. Based on *c*_1_ and *c*_2_, the THz-TDS system in reflection mode can be used to characterize the internal stress inside the TC layers of specimens.

#### 3.1.4. Calibration of Internal Stress of TBC

For the characterization of the stress inside the TC layers of specimens after thermal cycles, we collected THz spectra of all specimens. The average refractive index of S1, which was 4.1167, was used as the refractive index of the TC layer without stress. Based on Equation (5), the refractive indices and hence the stress of the TC layers of all specimens were obtained, as shown in Figure 8a, each result of which is the average of the seven different sampling spots. Figure 8b shows the detailed stress of different sampling spots for all specimens.

In Figure 8a, the average stress values of the TC layers of S2 to S7 were −1.0464 ± 0.0437 GPa, −0.9795 ± 0.0757 GPa, −1.1746 ± 0.0830 GPa, −1.2987 ± 0.0836 GPa, −1.3119 ± 0.0708 GPa and −1.3691 ± 0.0571 GPa, respectively. The TC layer was under the compressive stress state, and the value of average stress in the TC layer continuously increased with an increase in the number of thermal cycles. The internal stress evolution of EB-PVD TBC during the thermal cycle is similar to that of APS TBC in other related work [60,62]. In addition, rapid attenuation of stress was not observed in this work, indicating that no interfacial debonding occurred after 7000 thermal cycles. The stress increase rate was the fastest during the initial period of service and then gradually decreased. In addition, the standard deviation of the stress in the measurement area of the specimen increased continuously with increasing thermal cycles, indicating that the stress distribution in the TC layer became more uneven with the service time, as shown in Figure 8b.

### 3.2. Characterization of the Thermal Oxidation Stress Field of the TBC Specimen

#### 3.2.1. Theoretical Model of Cr^3+^ Photoluminescence Piezospectroscopy

The lattice deformation caused by the applied stress changes the transition energy between electronic or vibrational states, which is the basis of piezospectroscopic methods [63,64]. Cr^3+^ ions exist in *α*-Al_2_O_3_ corundum in the form of a trace impurity. After displacing some Al^3+^ ions with similar ionic radius, Cr^3+^ ions occupy the centre of normal octahedral ion sites. The frequency shift of the Cr^3+^ fluorescence peak occurs when *α*-Al_2_O_3_ corundum is subjected to compressive stress.

Grabner phenomenologically described the relationship between the frequency shift of fluorescence spectrum and the stress state [65]. The frequency shift, ∆*ν*, of the fluorescence spectrum can be expressed as follows:(6)$$\Delta v=\Pi_{ij}\sigma_{ij}{}^{*}=\left[\begin{array}{ccc} \Pi_{11} & \Pi_{12} & \Pi_{13}\\ \Pi_{21} & \Pi_{22} & \Pi_{23}\\ \Pi_{31} & \Pi_{32} & \Pi_{33} \end{array}\right]\left[\begin{array}{ccc} \sigma_{11}{}^{*} & \sigma_{12}{}^{*} & \sigma_{13}{}^{*}\\ \sigma_{21}{}^{*} & \sigma_{22}{}^{*} & \sigma_{21}{}^{*}\\ \sigma_{31}{}^{*} & \sigma_{32}{}^{*} & \sigma_{33}{}^{*} \end{array}\right]$$
where *σ*_ij_* is the stress component, *Π*_ij_ is the component of the wavenumber-stress coefficient tensor and the subscripts i and j indicate the crystallographic directions. Based on the point symmetry of Cr^3+^ ions, the *Π*_ij_ matrix can be reduced as [63]:(7)$$\Pi_{ij}=\left[\begin{array}{ccc} \Pi_{11} & 0 & 0\\ 0 & \Pi_{22} & 0\\ 0 & 0 & \Pi_{33} \end{array}\right]$$

Equation (6) can be expressed as:(8)$$\Delta v=\Pi_{11}\sigma_{11}{}^{*}+\Pi_{22}\sigma_{22}{}^{*}+\Pi_{33}\sigma_{33}{}^{*}$$

TGO conforms to the macroscopic isotropy hypothesis because its main component is polycrystalline *α*-Al_2_O_3_. There are enough randomly oriented grains in the scanning area. The Euler angle transformation matrix was used to express the spatial orientation of the piezospectroscopic relationship. When integrating over the entire space, the effect of the off-diagonal elements *Π*_ij_ (i ≠ j) and *σ*_ij_* (i ≠ j) disappears. Hence, Equation (8) can be expressed as:(9)$$\Delta v=\frac{1}{3}\left(\Pi_{11}+\Pi_{22}+\Pi_{33}\right)\left(\sigma_{11}+\sigma_{22}+\sigma_{33}\right)$$

Considering the film structure of TGO in TBC, the stress state of TGO can be simplified as the in-plane biaxial stress state (*σ*_11_ = *σ*_22_, *σ*_33_ = 0). Equation (9) can be reduced to:(10)$$\Delta v=\frac{2}{3}\left(\Pi_{11}+\Pi_{22}+\Pi_{33}\right)\sigma_{11}=\Pi_{e}\sigma_{11}$$
where *Π*_e_ is the piezospectroscopic coefficient under an in-plane equibiaxial stress state.

The thermal oxidation stress in TGO can be expressed as:(11)$$\sigma_{11}=\sigma_{22}=\frac{\Delta v}{\Pi_{e}}=\frac{\Delta v}{2\Pi_{U}}$$
where *Π*_U_ is the piezospectroscopic coefficient under a uniaxial stress state. The piezospectroscopic coefficients *Π*_U_ of TGO were 2.71 ± 0.04 cm^−1^/GPa and 2.50 ± 0.05 cm^−1^/GPa for R1 and R2, respectively [30].

#### 3.2.2. Characterization of Thermal Oxidation Stress

On the surfaces of each specimen, three sampling spots were randomly selected for the collection of Cr^3+^ photoluminescence spectra. In Figure 9a, the Cr^3+^ characteristic peak was not found in the Cr^3+^ photoluminescence spectra of S1, indicating that no TGO layer was formed in S1. In Figure 9b,c, we observed R1 and R2 peaks of Cr^3+^ photoluminescence spectra of S2 and S3, which meet the requirement of the characterization of the thermal oxidation stress.

Compared with the R2 peak, the R1 peak had a higher spectral intensity; hence, we characterized the thermal oxidation stress of all specimens using the R1 peak. In Figure 10 and Figure 11, the average TGO stresses of S2 to S7 were −2.79 ± 0.21 GPa, −3.54 ± 0.32 GPa, −3.98 ± 0.31 GPa, −4.23 ± 0.36 GPa, −4.17 ± 0.35 GPa and −4.49 ± 0.43 GPa, respectively. The average value of thermal oxidation stress continuously increased with an increase in the number of thermal cycles. At the beginning of the thermal cycle, the stresses increased at their fastest rate and then gradually stabilized. The stress evolution of the TGO was similar to that in the TC layer. Compared to APS TBC [34], the standard deviations of the TGO stresses of EB-PVD TBC are smaller, because the interface fluctuation of EB-PVD TBC is relatively small and gentle. The thermal oxidation stress field of S2 was generally uniform, showing a stress state of compressive stress. In Figure 10, the stress distribution became more and more haphazard with the increase in thermal cycle, which was further confirmed by the phenomenon in Figure 11, where the standard deviations of stress inside the measurement region increased continuously. In particular, after 2000, 5000 and 7000 thermal cycles, the value of stress obviously decreases at increasingly random positions, indicating that the stress was released owing to the initiation and expansion of interfacial cracks. In Figure 10, there is no large-scale stress release area as in other work [34,42], indicating that all specimens in this work still have a good thermal barrier coating structure.

### 3.3. Analyses and Discussions

Based on the THz-TDS and Cr^3+^-PLPS technologies, the evolution of stress inside the TC layer and thermal oxidation stress in the TGO of EB-PVD TBC during the long-term thermal cycling was characterized. As shown in Figure 8 and Figure 11, the internal and interfacial stresses of EB-PVD TBC had similar evolution rules. The evolution of stress is consistent with the parabolic law, which is consistent with the other relative literature [34,42]. For most specimens, the evolution of stress can be divided by two stages, I and II, corresponding to the initial stress increase and the stress stability.

The thermal oxidation stress in the TGO increased substantially during the first 1000 cycles due to the growth of the TGO, corresponding to stage I. The TGO gradually thickened and deformed at the TC/BC interface when the TBC was operated at high-temperatures, which in partly caused by the phase transformation of *θ*-Al_2_O_3_ to *α*-Al_2_O_3_. The change in its geometric configuration was constrained by the surrounding TC and BC layers, resulting in the TGO layer being under compressive stress. From 1000 to 7000 thermal cycles, the average values of the TGO stress were basically unchanged with increasing thermal cycles, corresponding to stage II. The growth of TGO stress almost stopped because the phase transformation of *θ*-Al_2_O_3_ to *α*-Al_2_O_3_ was basically finished by 1000 cycles [66]. Meanwhile, with the accumulation of interfacial stress, the initiation and expansion of cracks would be induced at the TC/BC interfaces, leading to the stress release. The competition between the two mechanisms led to the stabilization of the TGO stress during stage II.

The rapid increase in stress inside the TC layer during the early service process, corresponding to stage I, was caused by the phase transformation of the TC layer and the growth of the TGO layer. The metastable tetragonal (T’) phase, the main phase of the TC layer, was transformed into the tetragonal (T) phase and monoclinic (M) phase during the thermal cycles [67]. The significant volume expansion caused by the T’ to M phase transformation resulted in a compressive stress state inside the TC layer. Meanwhile, the growth and thickening of the TGO layer also affected the microstructure of the TC layer. The YSZ columns were pushed up (fanning of columns) by the TGO layer, leading to compressive stress in the TC layer. Similar to the phase transformation of the TGO layer, the phase transformation of the TC layer basically completed after the initial stage of service [68], leading to a decrease in the increase rate of stress inside the TC layer. Meanwhile, the accumulation of stress inside the TC layer led to the initiation and expansion of cracks and intensified the stress inhomogeneity. Hence, the stress inside the TC layer basically stabilized during stage II. In conclusion, the evolution of the internal and interfacial stresses of EB-PVD TBCs was interrelated and influenced by one another. Hence, the synergistic characterization of stress inside the TC layer and TGO stress of EB-PVD TBCs during long-term service is of great significance, and it is useful to explore the mechanism of stress evolution of EB-PVD TBCs.

As shown in Figure 8 and Figure 10, the stresses inside the TC layers of the specimens were less than those of the thermal oxidation stress values under the same number of thermal cycles. This was caused by the microstructure of the TC layer of EB-PVD TBCs [69], which is a columnar crystal microstructure. The columnar crystals were parallel to each other and were distributed perpendicular to the substrate. There were nanometre-scale pores between the ceramic columns [70], that is, intercolumn pores, whose orientations were basically perpendicular to the surface of the TC layer. The connection between the ceramic columns was weakened due to intercolumn pores. The stress inside the TC layer can be released through the intercolumn pore, thereby reducing the accumulated stress. Hence, the TC layer of the EB-PVD TBC had good toughness, thermal cycling performance and durability. In addition, as shown in Figure 1, peeling of the TC layer was not observed in S7, indicating the excellent thermal service performance of EB-PVD TBC. In particular, TBCs prepared using the EB-PVD technology have recently received attention due to their specific columnar structure, which may surpass APS TBCs in service performance.

## 4. Conclusions

To explore the evolutionary mechanism of the residual stress of EB-PVD TBCs, the characterization of the internal and interfacial stresses of the EB-PVD TBCs during the long-term service process was carried out. The stress-THz coefficients *c*_1_ and *c*_2_ of the EB-PVD TBCs were determined for the first time using a self-built, reflection-type THz-TDS system, which provided the information necessary for the stress-THz analysis of EB-PVD TBCs. After that, the internal and interfacial stresses of specimens during thermal cycles (from 100 to 7000) were characterized in combination with THz-TDS and Cr^3+^-PLPS analyses. Based on the experimental results, the evolution law of the stress inside the TC layer was similar to that of the TGO stress. They were interrelated and influenced by one another during the thermal cycling, which is meaningful for the exploration of the stress evolution mechanism of EB-PVD TBCs.

## Figures and Tables

**Figure 1 materials-16-02910-f001:**
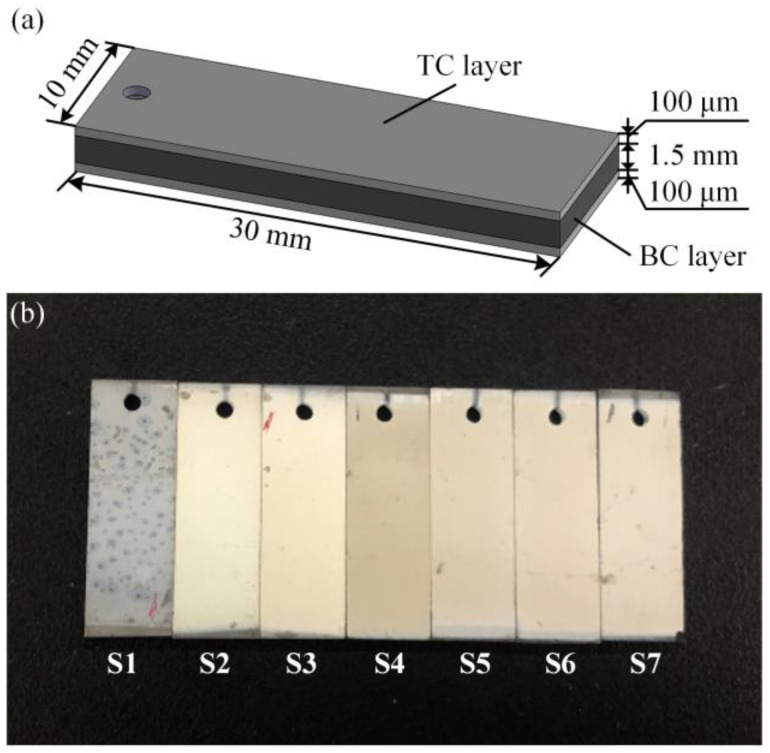
(**a**) Structure of the specimens used in this work; (**b**) specimens during thermal cycles.

**Figure 2 materials-16-02910-f002:**
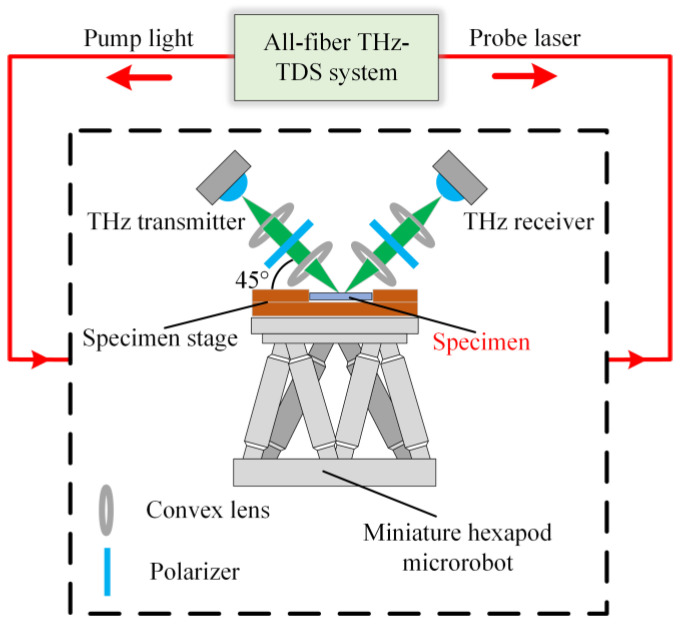
Schematic of the TBC internal stress characterization system.

**Figure 3 materials-16-02910-f003:**
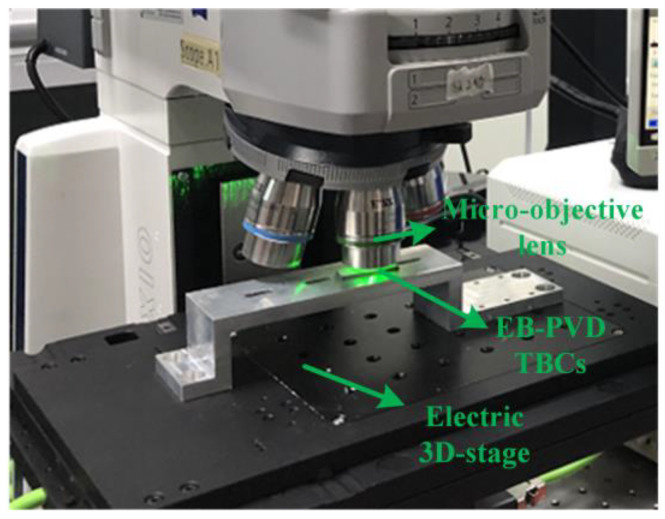
Schematic of the TGO stress characterization system.

**Figure 4 materials-16-02910-f004:**
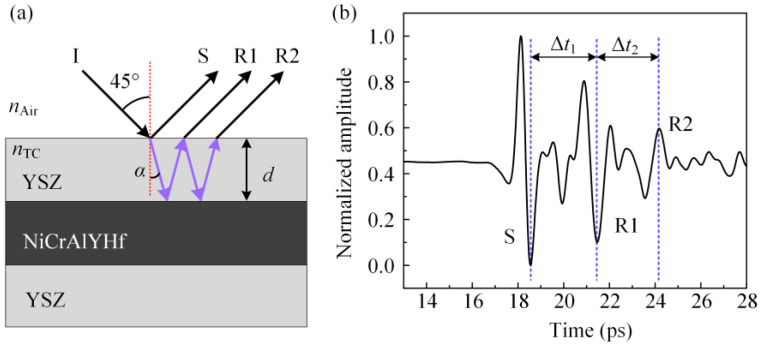
(**a**) The propagation path of a terahertz wave in specimens, (**b**) reflective terahertz time-domain spectrum.

**Figure 5 materials-16-02910-f005:**
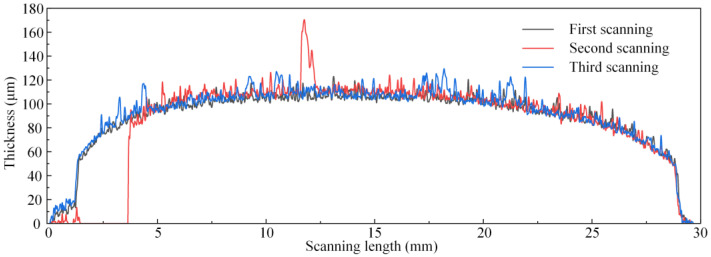
Measured results of surface topography scans of S1.

**Figure 6 materials-16-02910-f006:**
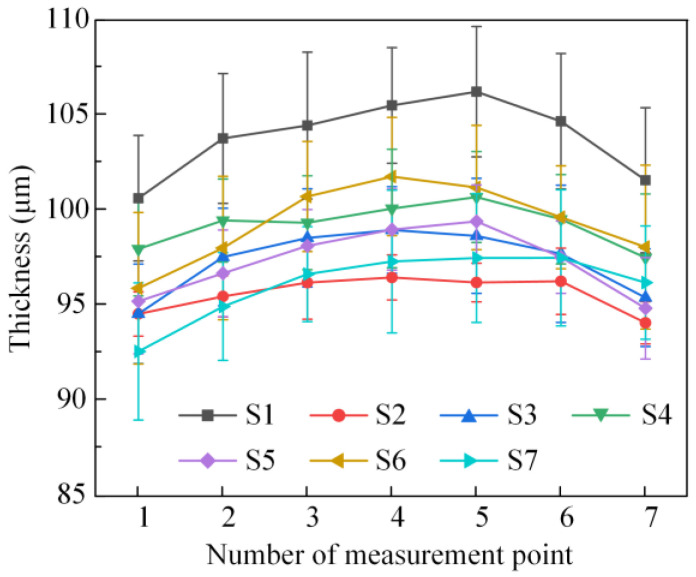
Measured thicknesses of the TC layers of the specimens after different numbers of thermal cycles.

**Figure 7 materials-16-02910-f007:**
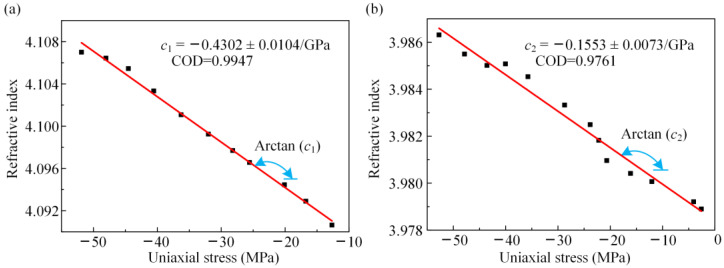
Results of calibration experiment: (**a**) stress-THz coefficients *c*_1_; (**b**) stress-THz coefficients *c*_2_.

**Figure 8 materials-16-02910-f008:**
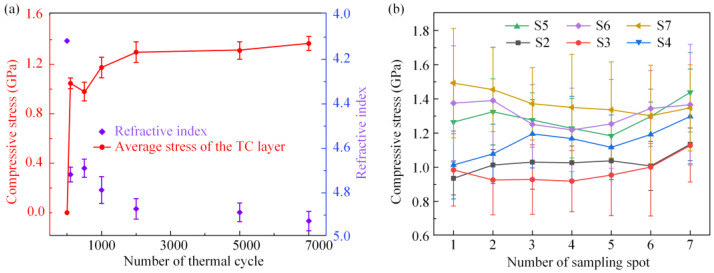
(**a**) Evolution of the average refractive indices and average stresses of the TC layers in the specimens after thermal cycles; (**b**) stress distributions of the TC layers in the specimens after thermal cycles.

**Figure 9 materials-16-02910-f009:**
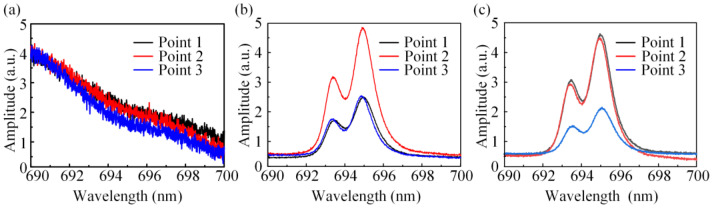
Cr^3+^ fluorescence spectra of specimens (**a**) S1, (**b**) S2 and (**c**) S3.

**Figure 10 materials-16-02910-f010:**
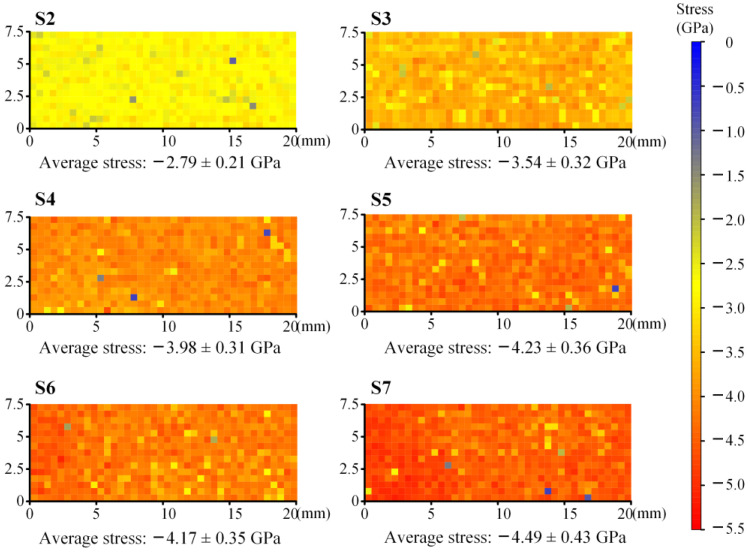
TGO stress fields of all specimens.

**Figure 11 materials-16-02910-f011:**
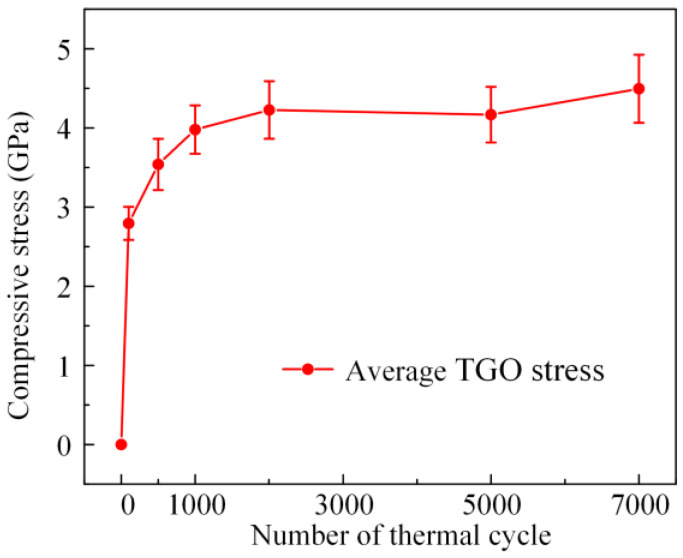
The average values of specimens after thermal cycles.

## Data Availability

The data are not publicly available because they also form part of an ongoing study.

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
