# Peer review of "Characterization of the Internal Stress Evolution of an EB-PVD Thermal Barrier Coating during a Long-Term Thermal Cycling"

_materials, 2023, doi:10.3390/ma16072910_

Round 1

Reviewer 1 Report

Comment 1: Qualitative informations are missing in abstract. Abstract should be concise and the authors need to improve with more specific short results.

Comment 2: Keywords is too, Keywords should be revised and improved.

Comment 3: The novelty of the work should be established.

Comment 4: Report the scattering bars in Figures 6 and 8(b). The following references should be added.

Journal of Molecular Liquids. 337 (2021) 116492 https://doi.org/10.1016/j.molliq.2021.116492

Journal of Molecular Liquids. 336 (2021) 116307 https://doi.org/10.1016/j.molliq.2021.116307

Comment 5: Compare your results with literature ones.

Comment 6: All variable must be italic but subscript or superscript symbols didn't italic. The author must be corrected all symbols of variables.

Comment 7: The number of equations should be mentioned in the text.

Comment 8: The introduction section should be modified though citing recent references (2022 and 2023) related studies and indicating the novelty of the study compared to the carried works.

Reviewer 2 Report

Generally, the work is good and the results are interesting. The research methods selected by the author are adequate for the subject of the paper. The topic is originally relevant in the field.

The conclusions are consistent with the evidence and arguments presented and they address the main question posed in the title of the publication.

The references are updated and consistent with the topic of work.

I recommend the article for publication in its present form.

Author Response

We appreciate the editors/reviewer for their earnest work and help in the revision of the manuscript. We would like to send our special thanks to you for your comments.

Reviewer 3 Report

The submitted manuscript discusses an experimental study on the internal stress of thermal barrier coatings (TBCs) prepared using electron beam physical vapor deposition (EB-PVD) technology. This is a standard industrial method for the preparation of a thermal barrier coating. The study presents mechanical models for stress measurements based on Cr3+ photoluminescence piezospectroscopy (PLPS) and terahertz time-domain spectroscopy (THz-TDS) to explore the mechanism of the stress inside EB-PVD TBCs. The results indicate that the internal stress inside the top coating layer and thermal oxidation stress were interrelated.

The presented topic is partially related to the emerging issues in aerospace and thermal power energy industries, there is an increasingly urgent demand for functional coatings. Overall, this study provides insights into the synergistic characterization of internal stress of EB-PVD TBCs during long-term thermal cycling.

The article should be revised as follows:

In the material preparation section, more information and characteristics on the used materials should be included (manufacturer, preparation, conditions, etc.) The authors should describe the thermal cycles in more detail and specify the temperature range.

The authors should check the typos, eg. line 115: “inlued”.

Reviewer 4 Report

The article deals with thermal coatings (TCB). This is a very relevant problem in science and in the production of spacecraft. The article is well written and logical. A lot of research has been done on this issue, but the novelty and originality of the results is clear. The article is proposed to be accepted for publication without corrections.

Author Response

We appreciate the reviewer for their earnest work and help in the revision of the manuscript. We would like to send our special thanks to you for your comments.

Reviewer 5 Report

Accept in the present form

Author Response

(The authors gave the same response as above.)

Reviewer 6 Report

The manuscript entitled Characterization of the Internal Stress Evolution of an EB-PVD Thermal Barrier Coating During a Long-term Thermal Cycling is an original article that reports experimental work on the stress evolution of EB-PVD TBCs during thermal cycling. 

The manuscript is well-organized, the research design is appropriate, the methods are accurately described and the results are clearly presented.  I recommend accepting the article after minor revision (text editing).

Author Response

(The authors gave the same response as above.)
